# The impact of hypoglycemia on quality of life and related outcomes in children and adolescents with type 1 diabetes: A systematic review

**Manon Coolen**[1]*, **Melanie Broadley**[1], **Christel Hendrieckx**[2,3], **Hannah Chatwin**[1], **Mark Clowes**[4], **Simon Heller**[5], **Bastiaan E. de Galan**[6,7,8], **Jane Speight**[1,2,3], **Frans Pouwer**[1,2,9], **for the Hypo-RESOLVE Consortium**[¶]

1 Department of Psychology, University of Southern Denmark, Odense, Denmark, 2 School of Psychology, Deakin University, Geelong, Victoria, Australia, 3 Australian Centre for Behavioural Research in Diabetes, Melbourne, Victoria, Australia, 4 Information Resources Group, School of Health and Related Research (ScHARR), University of Sheffield, Sheffield, United Kingdom, 5 Department of Oncology and Metabolism, University of Sheffield, Sheffield, United Kingdom, 6 Department of Internal Medicine, Maastricht University Medical Center+, Maastricht, The Netherlands, 7 CARIM School for Cardiovascular Diseases, Maastricht University, Maastricht, The Netherlands, 8 Department of Internal Medicine, Radboud University Medical Centre, Nijmegen, The Netherlands, 9 Steno Diabetes Center Odense, Odense, Denmark

¶ Membership of the HYPO-Resolve Consortium is provided in the Acknowledgments
* mcoolen@health.sdu.dk

**Data Availability Statement:** All relevant data are within the paper and its Supporting Information files.

## Abstract

### Objective

To conduct a systematic review to examine associations between hypoglycemia and quality of life (QoL) in children and adolescents with type 1 diabetes.

### Methods

Four databases (Medline, Cochrane Library, CINAHL, PsycINFO) were searched systematically in November 2019 and searches were updated in September 2021. Studies were eligible if they included children and/or adolescents with type 1 diabetes, reported on the association between hypoglycemia and QoL (or related outcomes), had a quantitative design, and were published in a peer-reviewed journal after 2000. A protocol was registered the International Prospective Register of Systematic Reviews (PROSPERO; CRD42020154023). Studies were evaluated using the Joanna Briggs Institute's critical appraisal tool. A narrative synthesis was conducted by outcome and hypoglycemia severity.

### Results

In total, 27 studies met inclusion criteria. No hypoglycemia-specific measures of QoL were identified. Evidence for an association between SH and (domains) of generic and diabetes-specific QoL was too limited to draw conclusions, due to heterogenous definitions and operationalizations of hypoglycemia and outcomes across studies. SH was associated with greater worry about hypoglycemia, but was not clearly associated with diabetes distress,

**Funding:** This work was supported by funding from the Innovative Medicines Initiative 2 Joint Undertaking (JU) (grant number 777460). The JU receives support from the European Union's Horizon 2020 research and innovation programme and EFPIA and T1D Exchange, JDRF, International Diabetes Federation (IDF), The Leona M. and Harry B. Helmsley Charitable Trust. CH and JS are supported by core funding to the Australian Centre for Behavioural Research in Diabetes provided by the collaboration between Diabetes Victoria and Deakin University. The funders had no role in study design, data collection and analysis, decision to publish, or preparation of the manuscript. URL: https://www.imi.europa.eu/.

**Competing interests:** The authors have declared that no competing interests exist.

depression, anxiety, disordered eating or posttraumatic stress disorder. Although limited, some evidence suggests that more recent, more frequent, or more severe episodes of hypoglycemia may be associated with adverse outcomes and that the context in which hypoglycemia takes places might be important in relation to its impact.

## Conclusions

There is insufficient evidence regarding the impact of hypoglycemia on QoL in children and adolescents with type 1 diabetes at this stage. There is a need for further research to examine this relationship, ideally using hypoglycemia-specific QoL measures.

## Introduction

Type 1 diabetes is one of the most common chronic conditions among children and adolescents and requires a demanding treatment regimen (e.g., insulin administration several times a day, monitoring of glucose levels and regulation of food intake and physical activity) [1, 2]. The goal of diabetes management is to achieve and maintain recommended glycemic levels to prevent/delay acute and long-term complications [1]. However, treatment with insulin can lead to hypoglycemia (low blood glucose level) [3]. Hypoglycemia can cause immediate uncomfortable symptoms (e.g., shakiness, dizziness), and in severe cases lead to confusion, seizures and coma, where self-treatment is not possible. In addition, recurrent episodes of severe hypoglycemia (SH) have been associated with neurocognitive impairments, especially in young children [4].

Although rates of SH in children and adolescents have decreased significantly in the past two decades, due to improvements in insulin administration and monitoring technologies (e.g., continuous subcutaneous insulin infusion and continuous glucose monitoring) [5–7], a recent systematic review still reported an incidence of 1.21–30 events per 100 person-years in young people with type 1 diabetes [8]. Hypoglycemia is particularly challenging and complex to manage in children and adolescents with type 1 diabetes for several reasons: this group has less predictable eating, activity, and sleep patterns relative to adults; children's diabetes is often (co-)managed by the parent; and young children may be unable to communicate their symptoms and needs [9]. Among adolescents, both hormonal changes leading to insulin resistance [10] and developmental changes, such as seeking independence from parents, that add to the burden of self-management, can lead to greater fluctuations in glucose levels and increase the risk of hypoglycemia [11].

Another important goal of pediatric diabetes management is to achieve and maintain optimal quality of life (QoL) [12]. While some studies have shown that hypoglycemia is negatively associated with QoL [13, 14], other studies have not found such an association [15, 16]. Although QoL is defined and assessed in many different ways across studies, it is recognized that QoL is a multidimensional, dynamic and subjective construct [17]. It has been argued that, to understand the impact of a condition on QoL, we need to ask people how satisfied they are with the areas of life that are important to them for their overall QoL, and then ask how these areas are affected by the condition, such as diabetes or, more specifically, hypoglycemia [18, 19]. It is therefore important to critically examine the range of patient-reported outcomes (PROs) used in studies, and to determine which are measuring the impact on QoL, and which are measuring related outcomes (such as diabetes-specific emotional distress or health status)

rather than QoL [18]. Synthesis of the current evidence base is needed to determine the relationship between hypoglycemia and QoL-related outcomes.

Therefore, our aim was to conduct a systematic review to summarize and critically appraise the evidence regarding the association between hypoglycemia and QoL (and related outcomes) in children and adolescents with type 1 diabetes.

## Methods

### Search strategy

This review was conducted in accordance with the Preferred Reporting Items for Systematic Reviews and Meta-Analyses (PRISMA) guidelines [20] and was registered on the International Prospective Register of Systematic Reviews (PROSPERO; CRD42020154023) database. A systematic search of Medline, Cochrane Library, CINAHL and PsycINFO databases was conducted in November 2019 and updated in September 2021, as part of a larger search strategy for five related systematic reviews examining the impact of hypoglycemia in various populations. Search terms included free-text and subject heading terms relating to the following concepts, separated by the Boolean operator "and": (1) type 1 diabetes, (2) children and adolescents, (3) hypoglycemia and (4) QoL and related outcomes. There were no limits applied to date or language at the search stage. The search string is provided in S2 File.

### Inclusion & exclusion criteria

Studies were eligible if they: (1) included children and/or adolescents with type 1 diabetes, majority aged ≤18 years (or mean age <18 years old), (2) assessed the history of hypoglycemia, (3) included outcomes of generic, diabetes-specific or hypoglycemia-specific QoL (or domains of QoL) or related outcomes (e.g., fear of hypoglycemia, depression, diabetes distress), (4) examined the association between hypoglycemia and QoL or related outcomes, (5) had a quantitative design, (6) were published in a peer-reviewed journal with full text available in English, (7) were published after 2000. The focus was limited to publications in the past two decades, as diabetes management strategies and rates of hypoglycemia have changed considerably in recent decades [5–7]. Studies were excluded if they: (1) focused on cognitive functioning [21] or neurodevelopmental disorders [22], or (2) only included proxy-report (e.g., by parents) of outcomes [23].

### Screening, article selection, and data extraction

Abstract screening was completed by three reviewers, with 10% of the abstracts being double screened (AS, AC and MCL). MC completed full text-screening (with input from a second reviewer (MB) where queries arose), and 10% of the full-text records were independently screened by a third reviewer (KM). In case of disagreement, reviewers discussed until consensus was reached. Additionally, forward chaining (i.e., citation searching of included studies in Google Scholar) and backward chaining (i.e., reference list checking of all included studies) was undertaken to identify additional eligible papers. Data extraction was performed by MC and KS; extracted data included reference details, study details, participant characteristics, analysis, results and discussion points. Extracted data were checked by two independent reviewers (HC, MB) and consensus was reached in case of discrepancies.

### Risk of bias assessment

Risk of bias was assessed (MC) by the analytical cross-sectional studies critical appraisal tool from the Joanna Briggs Institute (JBI) [24]. Risk of bias assessment was not used to exclude studies but was discussed and summarized to aid interpretation of the quality of the evidence base.

### Data synthesis

Narrative synthesis was structured primarily by a conceptual framework of QoL (Table 1), wherein outcomes were grouped based on two dimensions [18]: 1) the scope of the measure (global, broad, specific): i.e. whether a questionnaire assesses global QoL (e.g., 'overall QoL'), a broad domain of QoL (e.g., social functioning) or a specific domain of QoL (e.g., friends); and 2) the attribution of the measure (generic, diabetes- or hypoglycemia-specific): generic QoL measures ask people to rate areas of their life overall. These ratings can be affected by many factors including but also unrelated to diabetes, whereas diabetes-specific [or hypoglycemia-specific] QoL measures seek to attribute any impact to the condition, specifically asking: 'how does diabetes [or hypoglycemia] impact on your QoL?'

If a measure did not assess QoL but assessed a concept closely related to QoL (such as depressive symptoms, diabetes distress, FoH), it was classified as a related outcome. Diabetes distress refers to the negative emotions related to living with diabetes [25]. These related outcomes were grouped by the attribution of the measure (generic, diabetes- or hypoglycemia-specific outcome). Within each outcome group, study findings were summarized separately for SH and non-severe hypoglycemia (NSH) according to the authors' definitions, and where possible by outcome type/questionnaire.

To enhance consistency and transparency in the narrative synthesis and to avoid vote counting based on statistical significance [26], the following considerations were taken into account to interpret the evidence per outcome: 1) whether definitions and recall periods of hypoglycemia varied between studies; 2) whether there was a valid assessment of QoL and/or related outcomes; 3) whether analyses were conducted for children and adolescents separately or together; 4) whether there were exclusion criteria that directly related to hypoglycemia or QoL outcomes; and 5) whether effect sizes were available: small ($r \geq 0.10$ and $< 0.30$), moderate ($r \geq 0.30$ and $< 0.50$), and large $r \geq 0.50$) [27].

It was determined that there was insufficient evidence to draw a conclusion for an outcome if: a) there were less than three studies examining the association, or b) there was considerable heterogeneity in definitions of hypoglycemia and sample characteristics across studies.

## Results

### Included studies

The searches yielded 1165 results. Title and abstract screening resulted in 217 potential includes. After full-text screening, 17 studies were included. An overview of the full-text papers that have been assessed for eligibility with reasons for exclusion is provided in S1 Table. Forward and backward chaining yielded 10 extra includes. In total, 27 studies were included for data extraction and synthesis. Fig 1 provides an overview of the screening and selection process.

### Study characteristics

The 27 studies included a total of $N = 141,530$ participants, with sample sizes ranging from $N = 39$ to $N = 2,602$, with the exception of two large-scale studies ($N = 53,986$ and $N = 75,258$). The studies were conducted in 18 countries, the majority conducted in USA ($n = 6$) and Germany (n = 4). One study included multiple countries in Europe, North America and Japan [37]. The age of participants ranged from 5–25 years. Most studies included participants between 8–18 years, although eight studies also included participants above 18 years, and four studies also included participants aged 5–7 years. One study included children aged 6–12 years and five studies included adolescents aged 12–18 years. Study characteristics are detailed in Table 2.

**Table 1. Overview of quality of life and related outcome measures in the included studies, by breadth and attribution.**

| Quality of Life (QoL) and related outcomes | | Generic<br><br>(no attribution) | Diabetes-specific<br><br>(attribution to diabetes) | Hypoglycaemia-specific<br><br>(attribution to hypoglycaemia) |
|---|---|---|---|---|
| **Global QoL** | | • KINDL-R Total score [13] | • DISABKIDS DCGM-12 [13, 28] | None |
| | | • PedsQL total score [16, 29–31] | • DQOLY total score [32] | |
| | | | • DQOLY impact scale [32] | |
| | | | • DQOLY Diabetes life satisfaction scale [32] | |
| | | | • DQOLY Short Form total score [33] | |
| **Broad domains of QOL** | *Physical functioning* | • KIDSCREEN 27: physical wellbeing [14]<br>• KINDL R: physical [13, 34]<br>• PedsQL: physical functioning [16, 31]<br>• EQ5D VAS scale [14] | None | None |
| | *Social functioning* | • PedsQL: social functioning [31]<br>• PedsQL: psychosocial functioning [16, 31] | None | None |
| | *Psychological functioning* | • KIDSCREEN 27: psychological well-being [14]<br>• KINDL R: emotional wellbeing [13, 34]<br>• PedsQL: emotional functioning [31]<br>• KIDSCREEN-10 index [14] | None | None |
| **Specific domains of QoL** | *Family* | • KINDL R: family [13, 34]<br>• KIDSCREEN 27: Autonomy and relationships with parents [14] | None | None |
| | *Friends* | • KINDL R: friends [13, 34]<br>• KIDSCREEN 27: Relationships with friends or peers [14] | None | None |
| | *School / Studies* | • PedsQL: school functioning [31]<br>• KINDL R: school [13, 34]<br>• KIDSCREEN 27: school [14] | None | None |
| | *Self-esteem* | • KINDL R: self-esteem [13, 34] | None | None |
| | *Sleep* | • Adolescent Sleep/Wake scale [35] | None | None |
| **Related psychological outcomes** | | • Screen for Child Anxiety-Related Disorders [36]<br>• ICD-10 anxiety disorder diagnosis [40]<br><br>• State-Trait Anxiety Inventory for Children, Trait Subscale [38]<br><br>• Center for Epidemiological Studies-Depression Scale [43, 44]<br><br>• Children's Depression Inventory, Short version [48]<br>• Adolescents–IV (DSM IV depression diagnosis) [49] | • DQOLY: Worries about diabetes [32, 37]<br>• KINDL-R chronic illness scale [34]<br>• PedsQL DM: diabetes distress [15, 29, 30, 42, 45, 46]<br>• PedsQL DM: 'diabetes symptoms' [42]<br>• PedsQL DM: 'diabetes management' [42]<br>• DISABKIDS impact scale [13, 28]<br>• Diabetes Eating Problem Survey-Revised [50] | • Hypoglycaemia Fear Survey–child version (HFS-C) total scale [15, 38, 39]<br>• HFS-C 'worries about hypoglycaemia' subscale [36, 38, 41, 42]<br>• HFS-C 'fear of hypoglycaemia related behaviors'[36, 38, 41]<br>• Child Posttraumatic Stress<br>• Reaction Index (hypoglycaemia is referred to as the traumatic event) [47]<br>• Child Hypoglycaemia Index-2 [44] |

All studies had a cross-sectional design. Assessment of hypoglycemia relied mostly on retrospective self-report ($n$ = 19, 70%), with recall periods ranging from the past month ($n$ = 4) to the

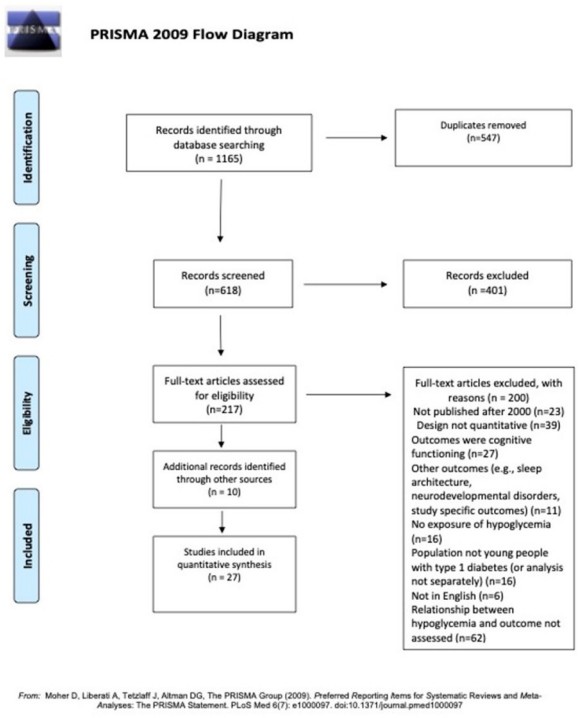

**Fig 1. PRISMA flowchart of the systematic search and screening, reasons for exclusions, and final number of included studies.**

past three ($n = 5$), six ($n = 5$) or 12 months ($n = 7$), to period since diagnosis ($n = 1$). Hypoglycemia was reported by the child or adolescent with diabetes ($n = 10$, 34%), their parent(s) ($n = 7$, 24%), or both ($n = 1$, 4%); or based on a combination of self-report and glucose meter data ($n = 2$, 8%). Six studies (22%) focused on hypoglycemia that involved coma or seizures, assessed via medical records. Two studies (7%) did not specify how hypoglycemia data were derived.

Twenty-one studies (78%) examined SH, defined as episodes: 1) where assistance of others was needed ($n = 4$); 2) resulting in confusion or seizures/coma ($n = 7$), or 3) characterized by a combination of these definitions ($n = 8$). Four studies did not specify a definition of SH. Ten studies (37%) examined "moderate" hypoglycemia, which will be referred to as "non-severe" hypoglycemia (NSH) throughout this review. Although it is recognized that NSH is usually referred to as "self-treated" hypoglycemia [51], that is not appropriate when describing hypoglycemia in (younger) children, as their parents often need to help regardless of the severity of the episode. Twelve studies (44%) used a continuous measurement of hypoglycemia (e.g., frequency of SH in the past 6 months), and fifteen (55%) reported on categorical measurement of hypoglycemia (e.g., absence or presence of SH).

The studies included 19 instruments to assess QoL and related outcomes (Table 1). The most commonly used were the Hypoglycemia Fear Survey Child Version (HFS-C) ($n = 6$) and the Pediatric Quality of Life Inventory (PedsQL) Generic ($n = 4$) and Diabetes ($n = 6$) modules. An overview of all scales being used in the studies can be found in S2 Table.

## Risk of bias assessment

Of the 27 studies, 93% included participants with a medically-verified diagnosis of type 1 diabetes. Most provided adequate details of their inclusion and exclusion criteria (89%) and

**Table 2.** Sociodemographic and clinical information and inclusion and exclusion criteria of the included studies.

| Author, year Country | Study design; Sample size | Age in years Mean (SD), Range | Diabetes duration in years Mean (SD) | Diabetes management | Inclusion/Exclusion criteria | HbA$_{1c}$ (DCCT unit) Mean (SD) | Hypoglycaemia assessments |
|---|---|---|---|---|---|---|---|
| Adler et al. (2017) [35] Israel | Cross-sectional N = 45 | 14.9 (1.7) R (12.2–17.9) | 5.9 (3.6) | MDI: 28.9% CSII: 71.1% CGM 35.6% | Included: age 6–30 years, diabetes duration: ≥1 year Excluded: psychiatric / neurological comorbidities, psychotropic medication, night shifts in the last 3 months, language difficulties | 7.96 (1.47) | No. of nocturnal H episodes last month: Less than once/week 48.9% 1–2 times/week 17.8% ≥3 times/week 6.7% |
| Al Hayek et al. (2014) [36] Saudi Arabia | Cross-sectional N = 187 | 15.3 (1.6) | 7.1 (5.2) | CSII 19.3% MDI 80.7% | Included: age 13–18 years, follow up for ≥12 months Excluded: psychopathological and medical instability, visual, hearing, or cognitive impairment | HbA$_{1c}$ >7 81.8% HbA$_{1c}$ ≤7 18.2% | Trouble with H past 12 months: 1–2 times: 7.5% 3–6 times: 34.9% 7–11 times: 16.6% ≥12 times: 41.8% Passed out due to H: 33.2% H episode while asleep: 82.9% H while awake but by themselves: 67.9% H in front of friends of strangers: 84% H when at school: 80.7% |
| Amiri et al. (2014) [39] Iran | Cross-sectional N = 61 | 9.2 (2.0) R (6.0–12.7) | 3.2 (2.0) R (0.5–10.5) | NR | Included: age 6–12 years, diabetes duration ≥6 months Excluded: other diseases (e.g., thyroid, celiac) | NR | Mean number of SH (past 3 months): 1.4 SD 5.4, range 0–36 |
| Caferoglu et al. (2016) [16] Turkey | Cross-sectional N = 70 | Median 13.0 R (11.00–15.00) | Median 3.5R (2.0–6.0) | MDI 100% | Included: aged 8–18 years, diabetes duration ≥1 year, using MDI Excluded: mental retardation and/or other chronic diseases (coeliac disease, hypothyroidism etc.) | Median 7.80, R (7.10–9.03) | Median and (Q1-Q3) number of NSH episodes 2.50 (0.00–5.25) |
| Coolen et al. (2021) [42] The Netherlands | Cross-sectional N = 96 | 15.2 (1.6) R 12–18 | 7.0 (4.3) | MDI:19% CSII: 81% CGM: 33% | Included: diabetes duration ≥ 6 months, no intellectual disabilities | 7.5 (.9) R 5.3–10.4 | No SH past 12 months: 80% SH past 12 months: 20% Mean number SH past 12 months: 0.7(2.4). Mean number of NSH past 6 months: 17.4 (29.9) |
| Dłużniak-Gołaska et al. (2019) [46] Poland | Cross-sectional N = 197 | 13.9 (2.3) R (8–18) | <5 years: 45.7% ≥5 years: 54.3% | CSII 100% CGM 31% | Included: diabetes duration ≥1 year, CSII treatment Excluded: other chronic diseases (e.g., coeliac disease) | NR | No/several times a month: 131 Several times a week/every day: 66 |
| Galler et al. (2021) [40] Germany, Austria, Switzerland, and Luxembourg | Observational N = 75,258 | 16.4 R 13.1–17.7 | 6.0 R 3.3–9.4 | CSII: 41% | Included: diabetes duration ≥1 year from 431 participating centers between 1995 until June 2019 | 7.9 R 7.1–9.0 | Rate of SH/patient year (95% CI): 12.8 (12.4; 13.3) |
| Gonder-Frederick et al. (2006) [38] USA | Cross-sectional N = 39 | 15.4 (1.5) | 7.0 (4.0) | CSII 36% | Included: age 12–17 years, diabetes duration ≥1 year Excluded: significant comorbidity (e.g., cystic fibrosis) and cognitive or learning disabilities | NR | Mean number NSH in past 12 months: 6.74, SD 5.03 Mean number SH past 12 months; 0.46, SD 2.11 |

*(Continued)*

**Table 2.** (*Continued*)

| Author, year Country | Study design; Sample size | Age in years Mean (SD), Range | Diabetes duration in years Mean (SD) | Diabetes management | Inclusion/Exclusion criteria | HbA$_{1c}$ (DCCT unit) Mean (SD) | Hypoglycaemia assessments |
|---|---|---|---|---|---|---|---|
| Hanberger et al. (2009) [28] Sweden | Cross-sectional N = 93 children N = 145 adolescents | 13.2 (3.9) R (8–19.6) | 5.1 (3.8) R (0.3–17.6) | CSII 17% | NR | 7.1(1.2) R (4.0–10.7) | NR |
| Hassan et al. (2017) [33] Egypt | Cross-sectional N = 150 | 12.3 (1.8) R (10–18) | <3 years: 46.7% 3–5 years: 34.7% >5 years: 18.6% | NR | Included: age 10–18 years, diabetes duration ≥1 year, completed diabetes education program | <7.5, 42.7% 7.5–9.0, 32% >9.0, 25.3% | SH with coma: 7% SH without coma: 93% |
| Hoey et al. (2001) [37] Multi country (17 countries in Europe, Japan and North America) | Cross-sectional N = 2101 | 13.8 R (10–18) | 5.2 | NR | Included: age 10–18 years, born between 1980–1987 | 8.7 (1.7) R (4.8–17.4) | Incidence of SH = 15.6 /100 patient years |
| Johnson et al. (2013) [15] Australia | Cross-sectional N = 196 | 11.8 (3.7) | 4.8 (3.5) | CSII 34.8% | Included: age 8–18 years old, diabetes duration ≥6 months, recent clinic attendance Excluded: significant comorbid condition, parent unable to answer the questionnaire | 8.0 (0.9) | SH: 18.8% |
| Jurgen et al. (2020) [44] USA | Cross-sectional N = 83 | 13.87 (3.21) | NR | CSII: 45% MDI: 24% 2 daily injections: 31% | Included: age 8–20 years, diabetes duration ≥1 year Excluded: type 2 diabetes, under 18 without parent, no HbA$_{1c}$ measurement, no blood glucose meter | 9.5 (1.8) | SH: 12.8% |
| Kalyva et al. (2011) [29] Greece | Cross-sectional N = 117 | 10.9 (4.0) R (5–18) | NR | MDI 99% CSII 1% | Included: age 5–18 years, diabetes duration ≥1 year | 8.05 (1.39) R (5.5–11.9) | Mean number of NSH episodes 5.82 SD 1.08, R 0–7 |
| Lawrence et al. (2012) [45] USA | Cross-sectional N = 2,602 | 13.6 (4.1) | 5.2 (3.9) | MDI 50% CSII 22% | Included: age >5 years, diabetes duration ≥1 year Excluded: not taking insulin, no HbA$_{1c}$ measurements | Good glycemic control, n = 32.3% Intermediate glycemic control, 47.6% Poor glycemic control = 20.1% | 0 SH = 88.1% 1 SH = 6.6% ≥ 2 SH = 5.3% |
| Matziou et al. (2010) [32] Greece | Cross-sectional N = 98 | 14.9 (2.4) | 7.3 (4.0) | CSII 32.7% | Included: age 11–18 years diabetes duration ≥6 months Excluded: psychiatric disorders | NR | NSH in past 3 months: 23.5% No NSH in past 3 months: 76.5% |
| Murillo et al. (2017) [14] Spain | Cross-sectional N = 136 | 13.5 (2.9) | 5.0 (3.7) | MDI 98.5% CSII 1.5% | Included: age 8–19 years, diabetes duration ≥6 months Excluded: cognitive problems | NR | SH in past 3 months: 2.2% No SH in past 3 months: 97.8% |
| Naughton et al. (2008) [31] USA | Cross-sectional N = 2,188 | 14.6 (3.6) | 6.2 (3.9) | Oral /no diabetes medications 0.6% MDI 76.9% CSII 22.5% | Included age≤20 years, resident in geographical center population, member of the participating health plan Excluded: diabetes as secondary to another condition | NR | 0 SH in past 6 months: 88.1% 1 SH in past 6 months: 6.4% ≥2 SH in past 6 months: 5.5% |

(*Continued*)

**Table 2.** (Continued)

| Author, year Country | Study design; Sample size | Age in years Mean (SD), Range | Diabetes duration in years Mean (SD) | Diabetes management | Inclusion/Exclusion criteria | HbA$_{1c}$ (DCCT unit) Mean (SD) | Hypoglycaemia assessments |
|---|---|---|---|---|---|---|---|
| Nip et al. (2019) [50] USA | Cross-sectional N = 2,156 | 17.7 (4.3) R (10–25) | NR | CSII 55% CGM 18.5% | Included: diabetes duration ≥5 years, diagnosed between 2002–2008 Excluded: type 2 diabetes not on insulin. | NR | NR |
| Plener et al. (2015) [49] Germany/ Austria | Observational N = 53,986 | NR | 5.77 | NR | Included: Age <25 years | NR | Rate of SH/patient year (95% CI)—Depression: 0.56 (0.52–0.58), No depression: 0.20 (0.19–0.20) Rate of SH coma/patient year (95% CI)—Depression: 0.04 (0.03–0.05), No depression: 0.03 (0.03–0.03) |
| Riaz et al. (2017) [43] Pakistan | Cross-sectional N = 104 | 15.8 (3.1) | 5.1 (4.0) | NR | Included: age 12–20 years, diabetes duration ≥1-year, recent clinic attendance Excluded: comorbid mental disorders or receiving psychotherapy | 10.3 (3.5) | SH in past six months = 20.2% |
| Serkel-Schrama et al. (2016) [30] The Netherlands | Cross-sectional online survey N = 129 | 14.0(2.0) R (12–18) | 6.0 (4.0) R (0–18) | CSII 71% | Included: age 12–18 years, self-reported type 1 diabetes, sufficient language skills | NR | No SH in last 12 months: 78% ≥1 SH last in 12 months: 12% |
| Shepard et al. (2014) [41] USA | Observational (validation study) N = 259[1] | 10.6 (3.3) R (6–18) | 5.2 (3.3) | MDI 60% CSII 40% | Included: diabetes duration ≥1 year, 4 BG readings/day for 4 weeks Excluded: medical comorbidities (e.g., asthma, cystic fibrosis), cognitive or learning disabilities | 8.01 (0.97) | NR |
| Sismanlar et al. (2012) [47] Italy | Cross-sectional N = 42 | M 13.67, SD 2.39 | 3.8 R (1–12) | NR | Included: age 8–18 years | 7.9 | SH: 28.6% H attacks in last month CTPS-RI<40: 7.11 (6.89), CPTS-RI ≥40: 13.57 (15.34) |
| Stahl-Pehe et al. (2013) [13] Germany | Cross-sectional N = 840 | M 16.3, SD 2.3 R (11.3–21.9) | M 13.3, SD 2.0, R (10.0–17.7) | CSII: 46.9% MDI 53.1% | Included: age 11–21 years, age of onset <5 years, diagnosed between 1993–1999, diabetes duration ≥10 years | 8.3 (1.4) R (5.6–15.4) | No SH in the last year: 41.7% SH in the last year (incl. last six months): 34.1% SH in the last month (incl. last week): 24.3% |
| Strudwick et al. (2005) [48] Australia | Cross-sectional N = 84 | 10.1 R (6–15) | 6.9 | NR | Included: age of onset <6 years, treatment at the center Excluded: neurologic or significant health problems unrelated to diabetes, psychiatric condition, developmental delay | NR | SH with seizures: 48.8% Number of SH: M; 2.5, SD; 2.2 |
| Wagner et al. (2005) [34] Germany | Cross-sectional N = 68 | 8–12 years: 72% 13–16 years 28% | M 4.2, SD 2.8, R (0.42–11.33) | MDI 100% | Included: age 8–16 years, diabetes duration ≥5 months | NR | SH: 19.6/100 patient years |

CGM, continuous glucose monitoring; CSII, continuous subcutaneous insulin infusion; H, hypoglycemia; MDI, Multiple daily injections; NR, not reported; SD, standard deviation; SH, severe hypoglycemia.

[a] Aggregation of five studies.

participants and settings (74%). In 52% of the studies, hypoglycemia was defined in accordance with the current International Society for Pediatric and Adolescent Diabetes (ISPAD) definition; namely, SH as an event with severe cognitive impairment (including coma and seizures) requiring assistance by others, and NSH as events with a blood glucose value ≤3.9 mmol/L (70 mg/dL) [52]. Most studies (81%) used statistical analyses appropriate to their data and, while all studies identified confounding factors, 67% adjusted analyses accordingly. Most studies (70%) used psychosocial outcome measures that were psychometrically valid and reliable instruments for use with children and adolescents with type 1 diabetes. In 11% of the studies, both validated and non-validated measures [13, 28, 36] were used, while 19% included measures that were not validated in adolescents with type 1 diabetes [14, 34, 35, 43, 47]. A full overview of the risk of bias assessment is presented in S3 Table.

## Narrative synthesis

Table 3 provides a summary of the main findings of each study.

**Global QoL.**  Table 1 shows that both generic measures and diabetes-specific measures of global QoL were used. One study used a non-validated questionnaire [13]. The age of the participants in these studies ranged from 11–21 years, and one study conducted analysis for children and adolescents separately [30].

*Severe hypoglycemia*. Three studies examined the relationship between SH and generic QoL using the KINDL-R [13] or the PedsQL [30, 31]. One study showed that those with SH (not further defined) in the past month reported significantly lower QoL than those without SH, but this was not observed for SH in the past year [13]. Two studies found no significant differences in generic QoL between groups with and without SH (not further defined [30] or episodes requiring assistance from others [31]) in the past 6 [31] to 12 months [30].

Three studies explored the association between SH and diabetes-specific QoL using the DISABKIDS DCGM-12 [13, 28] or the DQOL-Y [33]. In two studies, SH (not defined [13] or episodes requiring assistance [28]) in the past 12 months was not associated with diabetes-specific QoL after adjustment for covariates (e.g., age, gender, $HbA_{1c}$). However, one of these studies indicated that those who experienced SH (not further defined) in the past month reported significantly lower diabetes-specific QoL than those who had not [13]. The third study found that SH with coma since diagnosis was significantly associated with lower diabetes-specific QoL [33].

*Non-severe hypoglycemia*. One study found no significant differences diabetes-specific QoL scores on the DQOL-Y, between those who experienced NSH (glucose levels below 70 mg/dl) in the past three months and those who did not [32].

Two studies reported no significant association between frequency of NSH (glucose levels below 60 or 70 mg/dl) in the past month and generic QoL (PedsQL) after adjusting for covariates such as gender, hyperglycemia and age of onset of diabetes [16, 29].

**Broad and specific domains of QoL.**  Table 1 shows that studies examined three broad domains of QoL (psychological, physical, or social functioning), and/or the following specific domains of QoL: school, family, friends, self-esteem and sleep. Three studies used non-validated QoL questionnaires [13, 14, 34]. The participants' age range was 8–21 years, one study only included adolescents aged 12–18 [35].

*Severe hypoglycemia*. Four studies examined the relationship between SH (not further defined [13], or inability to self-treat due to neurological dysfunction [14, 34] or requiring assistance from others [31]) and broad domains (physical, psychological, social), or specific domains (self-esteem, family, friends or school) of QoL.

Three studies found no significant relationship between SH in the past 1–12 month(s) and physical functioning, [13, 14, 34]. In contrast, one study indicated that those with two or more

Table 3. Hypoglycaemia definition, measurement and relationship with quality of life and related outcomes.

| Author, year [ref] | Hypoglycaemia definition | Hypoglycaemia measurement | Recall period (months) | QoL domain or related outcome | Instrument | Findings: Association between hypoglycaemia and QoL / related outcome |
|---|---|---|---|---|---|---|
| Adler et al. (2017) [35] | Nocturnal H: BG levels <70 mg/dL or symptomatic H | No. nocturnal H episodes; self or parent reported | 1 | Sleep quality | ASWS | N.S. for sleep quality (data NR) |
| Al Hayek et al. (2014) [36] | Frequency of trouble with H episodes Passed out due to H H episode while asleep H episode while you were awake but by yourself H in front of friends or strangers? H when you were at school? | Categorical (1–2, 3–6, 7–11, >11) and yes vs. no; self-reported | 12 Ever | Worries about H; H related behavior; panic disorder; generalized anxiety disorder; separation anxiety disorder; social anxiety disorder; significant school avoidance | HFS -C SCARED | Pass out due to H associated with H related behaviors ($\beta = 0.502^{***}$), separation ($\beta = 0.189^{***}$) and school anxiety ($\beta = -0.271^{***}$) H while asleep associated with worries about H ($\beta = -0.508^{**}$) GAD ($\beta = -0.253$, p$^{**}$) and separation anxiety ($\beta = -0.274^{**}$) H while awake associated with H related behaviors ($\beta = -0.300^{*}$), worries about H ($\beta = -0.508^{**}$), panic disorder ($\beta = -0.318^{**}$), GAD ($\beta = -0.206^{**}$) and social anxiety ($\beta = -0.388^{***}$) H in front of friends associated with panic disorder ($\beta = 0.595^{***}$), GAD ($\beta = 0.537^{***}$), separation anxiety ($\beta = 0.321^{**}$), social anxiety ($\beta = 0.362^{**}$) and school anxiety ($\beta = 0.303^{***}$). H at school associated with H related behaviors ($\beta = -0.312^{*}$), panic disorder ($\beta = -0.284^{***}$), GAD ($\beta = -0.177^{*}$), separation anxiety ($\beta = -0.232^{**}$) and social anxiety ($\beta = -0.367^{***}$) All other associations are N.S. Covariates: age, gender, education, exercise, treatment type, duration of T1D, HbA$_{1c}$, passing out due to H, H as a big problem, H in front of friends and strangers and H at school |
| Amiri et al. (2014) [39] | SH: H with unconsciousness or consciousness but needing parent's help for treatment due to mental confusion and disorientation | No. of SH episodes; parent-reported | 3 | FoH Worries about H H related behaviors | HFS-C | N.S. for FoH (data NR) |
| Caferoglu et al. (2016) [16] | NSH: BG levels < 70 mg/dL, without seizures or coma | No. of NSH episodes; collected in interviews and checked with records from glucometers | 1 | Physical functioning; psychosocial functioning; general QoL | PedsQL | N.S. for psychosocial functioning, physical functioning and general QoL (p>0.05) |
| Coolen et al. (2021) [42] | SH: H when your blood glucose was so low that you were unable to recognize symptoms, ask for help, or treat yourself due to mental confusion or unconsciousness NSH: H when your blood glucose was so low that it interfered with what you were doing, and you had to wait a while to recover | No. of SH and NSH episodes; Self-reported | SH: 12 NSH: 6 | Worries about H DD Diabetes symptoms Diabetes management | HFS-C PedsQL DM | ↑ SH associated with ↑ worries about H$^{**}$ (r = 0.32) N.S. for NSH and worries about H (r = 0.17, p>0.05) N.S. for SH and NSH and diabetes distress, diabetes symptoms, or diabetes management (p>.05). Covariates include age, gender, HbA$_{1c}$, frequencies of H, perceived severity of H, fear of hypoglycaemia |

(Continued)

**Table 3.** (Continued)

| Author, year [ref] | Hypoglycemia definition | Hypoglycaemia measurement | Recall period (months) | QoL domain or related outcome | Instrument | Findings: Association between hypoglycaemia and QoL / related outcome |
|---|---|---|---|---|---|---|
| Dłużniak-Gołaska et al. (2019) [46] | NSH: BG levels < 70 mg/dL | No/several times a month vs. Several times a week/every day; self-reported | NR | DD | PedsQL- DM | N.S. for DD (p>0.05) Covariates: method of controlling glycemia, daily insulin dose, hyperglycemia, carbohydrate exchanges (CE) calculation and infections |
| Galler et al. (2021) [40] | SH: loss of consciousness or seizure or requiring assistance from another person to actively administer carbohydrates, glucagon, or intravenous glucose) | No. of SH episodes | NR | Anxiety disorders | ICD-10 German Modification | N.S. for rates of hypoglycaemia per 100 patient years between those with and without anxiety disorders (p>0.05) Covariates: age, sex, diabetes duration, migratory background, type of insulin therapy, and treatment year and depression |
| Gonder-Frederick et al. (2006) [38] | NSH: BG so low that it interfered with the adolescent's ability to function, but did not become so mentally disoriented that self-treatment was not possible SH: BG resulting in neuroglycopenia that interfered with the adolescent's ability to self-treat due to mental disorientation, unconsciousness, or seizure H in situations where the parent was not present (e.g., while sleeping, alone, at school, and in social situations) | No. of H episodes (severe and moderate); parent-reported | 12 | FoH Worries about H H related behaviors Trait Anxiety | HFS C STAIC | SH with unconsciousness ↑ FoH vs. no SH with unconsciousness* ↑ SH associated with ↑ worries about H** and FoH** Only for girls after adjustment for gender (Total: r = .59*; Worries: r = .55*) ↑ H episodes in social situations associated with ↑ trait anxiety (r = 0.37*) ↑ SH associated with ↑ FoH and ↑ worries about H** N.S. for SH and H related behaviors or trait anxiety (data NR) N.S. for H and FoH, worries about H, H related behaviors and trait anxiety (data NR) Covariates: trait anxiety scores frequency of H over the past year frequency of H, SH, episodes in situations where the child was likely alone) and gender |
| Hanberger et al. (2009) [28] | SH: needing assistance from another person | No SH vs. SH; self-reported | 12 | Diabetes-specific QoL and DD | DISABKIDS -DCGM-12 DISABKIDS Diabetes Module | N.S. differences for diabetes-specific QoL and DD (data NR) Covariates: gender, age, duration, HbA$_{1c}$ frequency of BG tests, parents living together or not, mother's educational level, use of insulin pump and center SH only associated with ↓ diabetes-specific QoL in single parent families, for adolescents (B = -1.22*) and children (B = -0.92*) |
| Hassan et al. (2017) [33] | SH with or without coma | SH with coma vs. SH without coma); taken from the medical record | NR | Diabetes-specific QoL | DQOL-Y | SH with coma vs. without coma associated with ↓ diabetes-specific QoL* |
| Hoey et al. 2001 [37] | SH: seizures or unconsciousness | No SH vs. ≥1 SH; self-reported | 3 | Worries about diabetes | DQOL-Y | ≥1SH associated with ↑ worries about diabetes than no SH (B = 4.2*) |

*(Continued)*

**Table 3.** (Continued)

| Author, year [ref] | Hypoglycaemia definition | Hypoglycaemia measurement | Recall period (months) | QoL domain or related outcome | Instrument | Findings: Association between hypoglycaemia and QoL / related outcome |
|---|---|---|---|---|---|---|
| Johnson et al. (2013) [15] | SH: event resulting in a seizure or coma | No SH vs. ≥1 SH; taken from Western Australia Childhood Diabetes Database | NR | DD FoH | PedsQL-DM HFS-C | N.S. for DD or FoH (p>0.05) Covariates: age and diabetes duration |
| Jurgen et al. (2020) [44] | SH: seizure or loss of consciousness | Parent reported | NR | FoH Depression | CHI-2 CES-DC | N.S. for FoH and depressive symptoms (r <.15, p>.05) |
| Kalyva et al. (2011) [29] | NSH: BG levels < 60 mg/dL without seizures or coma | No. of NSH episodes; parent-reported | 1 | General QoL DD | PedsQL PedsQL-DM | N.S. for general QoL or DD (p>0.05) Covariates: gender, age of onset episodes, number of hyperglycemic episodes, and HbA$_{1c}$ |
| Lawrence et al. (2012) [45] | SH: event requiring assistance of another person | No SH vs. 1 SH No SH vs. ≥ 2 SH; parent-reported | 6 | DD | PedsQL-DM | ≥ 2SH vs. no SH associated with ↑ DD** N.S. difference in no SH vs. 1 SH and DD (p>0.05) |
| Matziou et al. (2010) [32] | NSH: BG values <3.9 mmol/L (70 mg/ dL) | No NSH vs. ≥1 H; self-reported | 3 | Life satisfaction; disease impact; disease related worries; diabetes- specific QoL | DQOLY | N.S. for diabetes life satisfaction, disease impact, disease related worries and diabetes-specific-QoL (p>0.05) |
| Murillo et al. (2017) [14] | SH: BG levels <60 mg/dl with decreased level of consciousness requiring glucagon or the help of others | No SH vs. SH; taken from medical record | 3 | General QoL, health status, physical wellbeing; psychological wellbeing; parents/autonomy peers; school | EQ5D VAS KIDSCREEN-10 index KIDSCREEN 27 | SH vs. no SH associated with ↓ general QoL (ES 1.28*) N.S. for health status, physical wellbeing, psychological wellbeing, parents/autonomy, peers and school (p>0.05) |
| Naugthon et al. (2008) [31] | SH: event requiring assistance of another person | No SH vs. 1 SH No SH vs. ≥2 SH; self-reported | 6 | Overall generic QoL; psychosocial; social; school; physical health; emotional | PedsQL | ≥ 2 SH vs. no SH associated with ↓ physical health (β = -4.00***) N.S. for physical health between those with 1 SH vs. no SH and between those with 1 or ≥2 SH vs. no SH on general-QoL, social functioning, school functioning, emotional functioning and psychosocial functioning (p>0.05) Covariates: sex, race/ethnicity, age, highest level of parent education, and type of health insurance, BMI z score, duration of diabetes, type of diabetes treatment, HbA$_{1c}$ level, number of comorbid conditions, emergency department visits, and hospitalizations in the preceding 6 months |
| Nip et al. (2019) [50] | SH: event requiring assistance of another person | No. of SH; self-reported | 6 | Overall eating behavior | DEPS-R | N.S. difference in frequency of SH between those with DEB vs. without DEB (data NR) |
| Plener et al. (2015) [49] | SH with need of assistance of other persons, defined by unconsciousness, seizures, or application of glucagon or intravenous glucose | Rate of SH/patient year, rate of SH coma/patient year; taken from patient registries | H/patient year, recorded prospectively | Depression (diagnosis or symptoms) | ICD-10 and DSM-IV | SH /patient year ↑ in those with depression vs. without depression**N. S. for SH coma/patient year (p>0.05) |
| Riaz et al. (2017) [43] | SH and hospitalizations due to H | No SH vs. SH No hospitalization due to H vs. hospitalizations due to H | 6 | Depression | CES-D | N.S. for SH or hospitalizations due to H (p>0.05) |

*(Continued)*

**Table 3.** (Continued)

| Author, year [ref] | Hypoglycaemia definition | Hypoglycaemia measurement | Recall period (months) | QoL domain or related outcome | Instrument | Findings: Association between hypoglycaemia and QoL / related outcome |
|---|---|---|---|---|---|---|
| Serkel-Schrama et al. (2016) [30] | N/A | No SH vs. SH; parent-reported | 12 | General QoL and DD | PedsQL PedsQL-DM | SH vs. NO SH associated with ↑ DD (r = -0.19*) N.S. for generic QoL (p>0.05) |
| Shepard et al. (2014) [41] | SH (N/A) SH episodes requiring medical attention and NSH: % of readings <70 mg/dl | No. of SH episodes; Parent-reported | 12 | Helplessness; avoidance; maintaining high BG; social consequences | HFS C | ↑SH associated with ↑ helplessness (r = 0.19*) N.S. for SH and maintaining high BG, avoidance and worry about negative social consequences (data NR) Those who needed medical attention due to H vs. those without reported ↓ avoidance* N.S. for medical attention due to H and helplessness, maintain high BG and worry about negative social consequences (data NR) Children scoring in the highest tertile vs. the lowest tertile of maintain high BG had ↑SH episodes* N.S. for SH episodes, medical treatment due to H and % of readings <70 mg/dl and avoidance (p>0.05) and for medical treatment due to H and % of readings <70 mg/dl and maintaining high BG (p>0.05) |
| Sismanlar et al. (2012) [47] | NSH: BG levels <60mg/dl SH: H plus one of the following: BG levels ≤30 mg/dl, loss of consciousness, requirement of glucagon injection parenteral treatment at hospital | No. of SH; taken from BG charts and patients' home notes | 1 | PTSD | CPTS-RI | ↑SH associated with ↑ PTSD (β = 0.450*) N.S. for any SH and PTSD (data NR) N.S. difference in SH in last month or any SH between those with severe PTSD and those with mild/ moderate PTSD (p>0.05) |
| Stahl-Pehe et al. (2013) [13] | N/A | No SH vs. SH in past 12 months No SH vs. SH in past month; self-reported | 12 | Physical wellbeing; emotional wellbeing; self-esteem; family; friends; school; general QoL; diabetes impact; diabetes treatment; overall diabetes specific QoL | KINDL-R DISABKIDS | SH past year vs. no SH associated with ↓ quality of relationship friends (β = -3.1*) SH past month vs. no SH associated with ↓ emotional wellbeing (β =-4.2**), ↓ school functioning (β = -4.1*), ↓ general QoL (β = -3.0**), ↓ diabetes-specific QoL(β = -4.5**) ↓ diabetes impact (β = -3.8*) and ↓ diabetes treatment (β = -6.6**) N.S. association for SH past year or month vs. no SH and physical wellbeing, self-esteem, relationship with family (p>0.05) N.S. for SH past year and school functioning, diabetes-specific QoL, diabetes impact and diabetes treatment, general QoL emotional wellbeing (p>0.05) N.S. for SH past month and relationship with friends (p>0.05) Covariates: sex, age group, socioeconomic status, family structure, $HbA_{1c}$ level, insulin regimen, treatment satisfaction, weight status, and history of hospitalization |

(Continued)

**Table 3.** (Continued)

| Author, year [ref] | Hypoglycaemia definition | Hypoglycaemia measurement | Recall period (months) | QoL domain or related outcome | Instrument | Findings: Association between hypoglycaemia and QoL / related outcome |
|---|---|---|---|---|---|---|
| Strudwick et al. (2005) [48] | SH: resulting in seizure or coma | SH without seizure vs. SH with seizure; taken from medical record | Collected at clinics every 3 months | Depression | CDI -S | N.S. for depressive symptoms (data NR) |
| Wagner et al. (2005) [34] | SH: episodes with severe neurological dysfunction (e.g. seizures, loss of consciousness, disorientation, inability to arouse from sleep) that require intervention with glucagon or intravenous dextrose or milder forms of hypoglycaemia associated with neurological dysfunction that were not recognized or self-treated | No. of SH episodes | NR | Physical; psychological; wellbeing; self-esteem; family; friends; school, illness related distress | KINDL-R | N.S. for physical wellbeing, psychological wellbeing, self-esteem, family, friends, school and illness related distress (data NR) Covariates: age and gender |

ASWS, Adolescent Sleep Wake Scale; CDI-S, Children's Depression Inventory, Short version; CES-D, Center for Epidemiological Studies-Depression Scale; CHI-2, Child Hypoglycemia Index 2; CPTS-RI, Child Posttraumatic Stress Reaction Index, DEPS-R, Diabetes Eating Problem Survey-Revised; DSM-IV, Diagnostic and Statistical Manual of Mental Disorders; DQOLY, Diabetes Quality of Life for Youth scale; EQ-5D, EuroQoL 5 Dimensions; HFS-C, Hypoglycemia Fear Survey-Children version; ICD-10, International Classification of Diseases -10, PedsQL, Pediatric Quality of Life Inventory; PedsQL-DM, Pediatric Quality of Life Inventory-Diabetes Module; SCARED, Screen for Child Anxiety Related Emotional Disorders; STAIC, State-Trait Anxiety Inventory for Children. BG, blood glucose; DD, diabetes distress; FoH, fear of hypoglycemia; H, hypoglycemia; SH, severe hypoglycemia; No., number; N.S., Not significant (p>0.05); PTSD, Post-Traumatic Stress Disorder; QoL, quality of life, sig., significantly.

[a] Multivariate analysis are displayed only.

*$p<0.05$,

**$p<0.01$,

***$p<0.001$.

episodes of SH in the past six months reported significantly lower physical functioning than those without SH, but not for those who only had one SH [31].

Three studies found no significant relationship between SH in the past six [31] to 12 months and psychological functioning [13, 31, 34]. However, one study examined various recall periods and found that SH in the past month was significantly associated with lower psychological functioning [13]. The fourth study reported that those with SH in the past three months reported significantly lower psychological functioning than those without [14]. Two studies examined the association between in the past 1–12 months and self-esteem and found no significant results [13, 34].

One study reported no significant associations between SH in the past six months and social functioning [31]. Three studies reported no significant associations between SH in the past 1–12 months and relationships with family [13, 14, 34]. None of the studies reported a significant association between SH in the past 3–12 months and school [13, 14, 31, 34], although one study indicated that SH in the past month was significantly associated with lower school functioning [13]. Three studies examined the association between SH and quality of friendship [13, 14, 34]. One found that SH in the past year was significantly associated with lower quality of friendship, but this was not observed for SH in the past month [13]. Two studies found no significant association between SH and quality of friendship in the past 3–12 months [14, 34].

*Non-severe hypoglycemia*. Two studies examined associations of NSH and domains of QoL using the PedsQL [16] or the ASWS [35]. The first study found no significant associations between NSH (glucose levels below 70 mg/dl) in the past month and physical functioning or psychosocial functioning [16]. The second study found no association between nocturnal hypoglycemia (glucose levels below 70 mg/dl or symptomatic hypoglycemia) in the past month and adolescents' sleep quality [35].

**Related outcomes-hypoglycemia-specific.** Table 1 shows that although hypoglycemia-specific QoL was not assessed, related outcomes were assessed, including fear of hypoglycemia (FoH), and hypoglycemia-specific post-traumatic stress symptoms. One study used a non-validated questionnaire [47]. The participants' age range was 6–20 years. Three studies specifically focused on adolescents aged 12–18 [36, 38, 42] and one on children aged 6–12 [39].

*Severe hypoglycemia*. Seven studies examined the association between SH and FoH measured with the HFS-C [15, 36, 38, 39, 41, 42] or the CHI-2 [44].

Four studies examined relationships between SH and worries about hypoglycemia [36, 38, 41, 42] Three of these studies reported significant, small-to-medium, positive correlations between frequency of SH (not further defined [41] or inability to self-treat due to mental disorientation or seizures [38, 42]) in the past 12 months and greater worries about hypoglycemia. However, in one study, this only remained statistically significant for female adolescents after controlling for gender [38]. The fourth study found no significant difference in worries about hypoglycemia between adolescents who never lost consciousness and those who ever lost consciousness due to SH, after controlling for covariates such as age gender and other types of hypoglycemia [36].

Three of the studies also explored associations between SH and FoH related behaviors [36, 38, 41]. One study reported no significant association between frequencies of SH episodes (inability to self-treat due to mental disorientation or seizures) and FoH related behaviors [38]. The second study indicated that those who had passed out due to hypoglycemia significantly reported more hypoglycemia related avoidance behaviors compared to those who had never passed out [36]. In contrast, the third study reported that those who needed medical attention due to SH reported significantly less hypoglycemia related avoidance behaviors than those who did not [41].

Three of the eight studies examined associations between SH (inability to treat due to mental confusion or unconsciousness in the past three months [39] or SH resulting in seizures or coma [15, 44]) and overall FoH, but found no significant associations [15, 39, 44].

An additional study found that frequency of SH (loss of consciousness or requirement of glucagon) in the past month was a significant predictor of self-reported post-traumatic stress (PTSD) assessed with the CPTS-RI after adjustment for age and family history of diabetes [47].

*Non-severe hypoglycemia.* Four studies explored the relationship between NSH and HFS-C subscale scores [36, 38, 41, 42]. Three of these studies reported no significant associations between frequency of NSH (glucose values below 70 mg/dl [41] or interfering with ability to function [38, 42]) and FoH. The fourth study found that frequency of NSH and 'hypoglycemia while at school', 'awake' or 'asleep' were significantly associated with at least one of the HFS-C scales, after adjustment for clinical factors and other types of hypoglycemia (e.g., passing out because of hypoglycemia). This was not observed for 'hypoglycemia in front of friends' [36].

**Related outcomes—diabetes-specific.** Table 1 shows that studies assessed the relationship between hypoglycemia worries attributed to diabetes, diabetes-related disordered eating and diabetes distress. One study used a non-validated questionnaire [34]. The participants' age range was 5–21 years. Two studies focused on adolescents aged 12–18 [30, 42] and one study conducted analysis for children and adolescents separately [28].

*Severe hypoglycemia.* Eight studies investigated the association between SH and diabetes distress [13, 15, 28, 30, 34, 37, 42, 45]. Four of these studies used the PedsQL DM [15, 30, 42, 45]. One of these reported that SH (not further specified) in the past 12 months was significantly associated with higher diabetes distress, compared to those without SH, with a small effect size [30]. This was confirmed for those who experienced two or more SH episodes (requiring assistance from others) in the past six months, but not for only one SH [45]. In contrast the other two studies reported no significant association between SH (inability to self-treat due to mental confusion) in the past 12 months) [42] or SH (resulting in seizure or coma) since diagnosis [15] and diabetes distress.

Two of the studies using the DISABKIDS Diabetes Module found a significant association between SH (not further defined [13] or requiring assistance from others [28]) and greater diabetes distress if hypoglycemia was experienced in the past month [13] but not in the past year [13, 28]. One study reported no significant association between SH (inability to self-treat due to neurological dysfunction) and illness-related distress using the KINDL-R [34].

An additional study (using the DQOL-Y) reported that those who had SH involving seizures or coma in the past three months reported significantly more worries about diabetes than those without [37].

In addition, in one study frequency of SH (requiring assistance from others) in the past 6 months did not significantly differ between those with and without disordered eating measured with the DEPS-R [50].

*Non-severe hypoglycemia.* Four studies examined the association between NSH (glucose concentrations below 60 or 70 mg/dl [29, 32, 46] or interfering with ability to function [42]) in the past 1–6 months and diabetes distress using the PedsQL DM [29, 42, 46] or the DQOL-Y [32]. None of these studies reported a significant association between NSH and diabetes distress.

**Related outcomes–generic.** Table 1 shows that generic outcomes including anxiety or depression symptoms, or diagnosis were examined in the studies. Some of the studies used measures that are not validated in young people with diabetes [36, 38, 43, 44]. The participants' age range was 0–25 years.

*Severe hypoglycemia.* Three studies reported no significant association between SH and depressive symptoms; the first explored the association between SH in the past 6 months (not further defined) and CDI-S scores [48] and the other two explored the association between SH resulting in coma or seizure and CES-D scores [43, 44]. In contrast, another study reported a significant positive relationship between SH (requiring assistance from others and unconsciousness or application of glucagon) in the past year and a DSM-IV depression diagnosis [49].

Two studies investigated the associations between SH and anxiety symptoms assessed by the SCARED [36] or an ICD-10 anxiety disorder diagnosis [40]. The first study reported that a history of passing out due to SH was significantly associated with greater symptoms of separation anxiety and school avoidance, but not with panic disorder, generalized anxiety or social anxiety [36]. The second study reported no significant associations between SH (loss of consciousness) and diagnosis of anxiety disorders [40].

*Non-severe hypoglycemia.* Two studies examining associations between hypoglycemia in different situations and various anxiety types (using the SCARED [36] or STAIC [38]) found that having 'hypoglycemia while at school', 'in front of strangers', 'while awake' or 'asleep' [36] was significantly associated with greater symptoms anxiety, for example social anxiety or separation anxiety [36] and that hypoglycemia in social situations was significantly associated with higher trait anxiety, with a moderate effect size [38].

## Discussion

To our knowledge, this is the first systematic review that critically examines evidence on the relationship between hypoglycemia and QoL and related outcomes among children and adolescents with type 1 diabetes. Results of this review show that evidence regarding an association between SH and (domains of) generic QoL is inconclusive, while the evidence suggests no association between NSH and generic QoL. For diabetes-specific QoL, the evidence was too limited to draw conclusions. None of the studies used hypoglycemia-specific QoL measures to explore the association between hypoglycemia and QoL. In addition, there was some evidence suggesting an association between SH in the past 12 months and greater worries about hypoglycemia, and no association between NSH and diabetes distress. There was insufficient evidence to draw conclusions regarding the relationship between hypoglycemia and diabetes distress (for SH), FoH worries (for NSH), FoH-related behaviors and total FoH, anxiety, depression, disordered eating and PTSD.

A possible explanation for inconsistent findings is the heterogeneity in definitions of and recall periods for hypoglycemia and measures used to assess QoL across studies. This variation limits the ability to compare studies and draw conclusions. Several key limitations of the existing evidence base were identified, such as cross-sectional designs, low statistical power, lack of reporting of effect sizes (and thus limited information on the clinical value of the observed statistically significant differences), lack of information on the definition or frequency of hypoglycemia, and the self-report of hypoglycemia over several months or even back to diagnosis, which might have led to recall bias. The key recommendation for future studies is to use a definition of hypoglycemia as recommended by current guidelines. Future studies should also use longitudinal /prospective study designs and modern methods, such as continuous glucose monitoring, for a more objective assessment of hypoglycemia that does not rely on recall of episodes, to determine the direct, day-to-day impact of hypoglycemia on various domains of QoL in children and adolescents with type 1 diabetes.

Although the current evidence suggests no clear association between hypoglycemia and some outcomes, it is important to note that studies were more likely to show statistically significant associations between hypoglycemia and outcomes when SH was experienced recently (in the past 1–3 months), more frequently, or when it involved convulsions, unconsciousness, or coma. In addition, some studies suggested that the context in which hypoglycemia takes places (e.g., in social situations) might have implications for its impact. However, this was only based on a few studies, some of which have methodological limitations. Thus, more evidence is needed to confirm these associations.

Although emerging evidence shows the importance of self-treated hypoglycemia in relation to QoL and related outcomes in adults with diabetes (50, 51), current evidence on this

relationship in youth with type 1 diabetes suggested no association between NSH and QoL. However, this should be interpreted with caution, as these studies are limited by the use of generic and diabetes-specific QoL questionnaires, while hypoglycemia-specific QoL measures may be more sensitive to the impact of NSH. Different research designs, that minimize recall bias and assess the impact closer to the occurrence NSH are needed to understand the association between NSH and QoL. In addition, this review identified only one study that explored the association between hypoglycemia while asleep and sleep quality (28). This highlights the need for more studies that investigate such relationships.

Although QoL has been considered as a key outcome in pediatric diabetes care [12], only two studies had a primary aim to examine the impact of hypoglycemia on QoL [15, 42]. Fifteen of the 27 studies aimed to explore QoL, however, only seven of these studies included measures that actually assess QoL, whereas the others focused on particular domains of QoL or measured related outcomes such as diabetes distress [13, 15, 28, 30, 34, 37, 42, 45] or health status [14], rather than QoL. Even though other studies included in this review have focused on identifying sociodemographic and clinical factors that are associated with QoL, it is difficult to identify the impact of hypoglycemia specifically in these studies [13, 14, 16, 28, 30–34, 37, 45, 46]. Importantly, some of the studies that explored the impact of hypoglycemia on QoL as a secondary aim had very low rates of hypoglycemia in their samples. To truly understand the impact of hypoglycemia on QoL, a questionnaire that assesses how hypoglycemia affects domains of life that are important to the individual should be used [18]. Given that there are currently no hypoglycemia-specific QoL measures that are designed to assess the impact of hypoglycemia on QoL in children and adolescents with type 1 diabetes, these need to be developed and would need to be age appropriate and to incorporate specific domains that are important to young people with diabetes. There might be other domains of importance to young people's QoL, such as leisure activities, that were not included in questionnaires used in current studies.

Finally, the majority of studies included in this review pooled children and adolescents together when examining the link between hypoglycemia and QoL or related outcomes. Although these studies usually included age-appropriate assessments of outcomes, the impact of hypoglycemia on these outcomes might be different for children and adolescents. While younger children often rely on their parents for decisions about diabetes management, these responsibilities are usually transferred to the child during adolescence [53, 54]. During this challenging process of transferring responsibilities, the burden of self-management for the adolescents increases and can lead to increased hypoglycemia, which can interfere with other demands and lead to family conflicts, reduced self-efficacy and increased FoH, all aspects that can compromise QoL in adolescents with diabetes [42, 55]. Future studies should thus explore if the relationship between hypoglycemia and QoL is different in different age groups. Additionally, adolescence is characterized by a strong desire to be accepted by peers [54]. Episodes of hypoglycemia in this age group could be experienced as embarrassing and mark out adolescents with diabetes as different. Hypoglycemia has indeed previously been associated with higher stigma in young people with diabetes [56], which can lead to poorer psychosocial and medical outcomes [57]. Future studies need to explore the role of stigma as a possible mechanism by which hypoglycemia impacts on QoL.

## Strengths and limitations

Strengths of this review include the systematic and comprehensive search of multiple databases, and the application of a conceptual framework of QoL to categorize outcome measures in order to critically appraise the evidence. This allows for a more detailed understanding of the various ways in which hypoglycemia can impact on QoL and related outcomes and

highlights the gaps in the evidence base. This review also has some limitations. Although the inclusion of a wide range of outcomes provided an overview of all the available evidence related to the impact of hypoglycemia on QoL, it also made it difficult to compare studies directly. Further, the heterogeneity across studies and the lack of effect sizes reported in studies, precluded the possibility of meta-analysis. The inclusion of only quantitative studies that were published in English may have introduced some bias, although only six studies were excluded for this reason.

## Implications for clinical practice

The implications for clinical practice that can be drawn from this review are limited due to the inconclusive and relatively small evidence-base. However, some evidence suggests that more recent episodes of hypoglycemia might have an impact on various outcomes. This may be useful for clinicians, as they could ask specifically about hypoglycemia and its impact in the weeks/months following episodes of SH.

## Conclusion

This systematic review shows that there is insufficient evidence on the relationship between hypoglycemia and (domains of) generic and diabetes-specific QoL in children and adolescents with type 1 diabetes. This is largely because heterogeneity and methodological limitations across studies hamper the ability to draw strong conclusions. Importantly, none of the studies used a measure designed specifically to assess the impact of hypoglycemia on QoL. Additionally, there seems to be an association between SH and greater worry about hypoglycemia, while the evidence is too limited for other related outcomes. Although limited, some evidence suggests that issues such as timing and context of hypoglycemia might influence its impact. Future research should focus on the development of measures that can assess the impact of hypoglycemia in children and adolescents with type 1 diabetes and use agreed definitions of hypoglycemia that increase comparability between studies.

## Supporting information

**S1 File. Protocol as registered on PROSPERO.**
(PDF)

**S2 File. Full search strategy.**
(DOCX)

**S1 Table. Overview of full text papers assessed for eligibility with reasons for exclusion.**
(DOCX)

**S2 Table. Overview of scales being used across studies.**
(DOCX)

**S3 Table. Quality assessment of the included studies.**
(DOCX)

**S1 Checklist.**
(DOCX)

## Acknowledgments

The authors thank Helen Buckley-Woods (ScHARR) for running the search and Louise Preston (ScHARR) for her work on the data extraction form, and Anna Cantrell (ScHARR) and

Anthea Sutton (ScHARR) for abstract screening, and Katie Sworn (ScHARR) for data extraction, and Kevin Matlock (University of Southern Denmark) for double-screening 10% of the full-text records. In addition, individual authors and affiliations within the Hypo-RESOLVE Consortium are listed in the table below.

| Participant organisation name | Scientific person(s) in charge |
|---|---|
| STICHTING RADBOUD UNIVERSITAIR MEDISCH CENTRUM | Prof. Cees Tack |
| | Dr Bastiaan de Galan |
| King's College London | Prof. Stephanie Amiel |
| | Dr Pratik Choudhary |
| Medical University of Graz | Prof. Thomas Pieber |
| | Dr Julia Mader |
| University of Cambridge | Dr Mark Evans |
| Montpellier University Hospital | Prof. Eric Renard |
| University of Southern Denmark | Prof. Frans Pouwer |
| | Prof. Jane Speight |
| University of Lausanne | Prof. Bernard Thorens |
| University of Sheffield | Prof. Simon Heller |
| | Prof. Alan Brennan |
| Nordsjællands University Hospital Hillerød | Prof. Ulrik Pedersen-Bjergaard |
| University of Dundee | Prof. Rory McCrimmon |
| European Research and Project Office GmbH | Jakob Haardt |
| Swiss Institute of Bioinformatics | Dr Mark Ibberson |
| University of Padova | Prof. Giovanni Sparacino |
| University of Edinburgh | Prof. Helen Colhoun |
| Novo Nordisk A/S | Dr Stephen Gough |
| Eli Lilly and Company Limited | Dr Zvonko Milicevic |
| Abbott Laboratories | Dr Mahmood Kazemi |
| Medtronic International Trading Sàrl | Dr Ohad Cohen |
| JDRF International | Dr Sanjoy Dutta |
| International Diabetes Federation | Dominique Robert |
| Unitio, Inc. | Dr Wendy Wolf |
| The Leona M. and Harry B. Helmsley Charitable Trust | Dr Sean Sullivan |

## Author Contributions

**Conceptualization:** Manon Coolen, Melanie Broadley, Christel Hendrieckx, Simon Heller, Bastiaan E. de Galan, Jane Speight, Frans Pouwer.

**Data curation:** Manon Coolen, Melanie Broadley, Hannah Chatwin, Mark Clowes.

**Formal analysis:** Manon Coolen, Melanie Broadley, Hannah Chatwin, Jane Speight, Frans Pouwer.

**Funding acquisition:** Christel Hendrieckx, Simon Heller, Bastiaan E. de Galan, Jane Speight, Frans Pouwer.

**Investigation:** Manon Coolen, Melanie Broadley, Hannah Chatwin, Mark Clowes.

**Methodology:** Manon Coolen, Melanie Broadley, Jane Speight, Frans Pouwer.

**Project administration:** Melanie Broadley, Jane Speight, Frans Pouwer.

**Supervision:** Melanie Broadley, Jane Speight, Frans Pouwer.

**Validation:** Manon Coolen, Melanie Broadley, Hannah Chatwin, Mark Clowes.

**Writing – original draft:** Manon Coolen.

**Writing – review & editing:** Melanie Broadley, Christel Hendrieckx, Hannah Chatwin, Mark Clowes, Simon Heller, Bastiaan E. de Galan, Jane Speight, Frans Pouwer.

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
