## [Decision Letter · Decision Letter 0]

13 Sep 2021

PONE-D-21-24300The impact of hypoglycemia on quality of life and related outcomes in children and adolescents with type 1 diabetes: a systematic reviewPLOS ONE

Dear Dr. Coolen,

Thank you for submitting your manuscript to PLOS ONE. After careful consideration, we feel that it has merit but does not fully meet PLOS ONE’s publication criteria as it currently stands. Therefore, we invite you to submit a revised version of the manuscript that addresses the points raised during the review process.

Besides the important reviewers' comments provided below, authors are encouraged to address the following:

Date of search was November 2019 (about 2 years ago); authors should update their review to include possible recent publications in 2020 and 2021, such as (Gordon, J., Beresford-Hulme, L., Bennett, H., Tank, A., Edmonds, C., & McEwan, P. (2020). Relationship between hypoglycaemia, body mass index and quality of life among patients with type 1 diabetes: Observations from the DEPICT clinical trial programme. *Diabetes, obesity & metabolism*, *22*(5), 857–865) and (Coolen M, Aalders J, Broadley M, Aanstoot HJ, Hartman E, Hendrieckx C, Nefs G, Pouwer F. Hypoglycaemia and diabetes-specific quality of life in adolescents with type 1 diabetes. Diabet Med. 2021 Aug;38(8):e14565).Authors should provide a list of publications assessed for eligibility, with reasons for exclusion of many of them.

We look forward to receiving your revised manuscript.

Kind regards,

Elsayed Abdelkreem, MD, PhD

Academic Editor

PLOS ONE

Journal Requirements:

3. One of the noted authors is a group or consortium Hypo-RESOLVE Consortium. In addition to naming the author group, please list the individual authors and affiliations within this group in the acknowledgments section of your manuscript. Please also indicate clearly a lead author for this group along with a contact email address.

Reviewers' comments:

Reviewer's Responses to Questions

**Comments to the Author**

1. Is the manuscript technically sound, and do the data support the conclusions?

Reviewer #1: Yes

Reviewer #2: Partly

2. Has the statistical analysis been performed appropriately and rigorously? 

Reviewer #1: I Don't Know

Reviewer #2: N/A

3. Have the authors made all data underlying the findings in their manuscript fully available?

Reviewer #1: Yes

Reviewer #2: Yes

4. Is the manuscript presented in an intelligible fashion and written in standard English?

Reviewer #1: Yes

Reviewer #2: No

5. Review Comments to the Author

Reviewer #1: In data analysis , some terms needs to be explained as "diabetes distress"

Regarding the conclusion of the study , why did the authors mentioned that the was insufficient evidence on the relationship between hypoglycemia and QOL despite that we can see from the studies involved in the meta- analysis systematic review that QOL was significantly affected in hypoglycemia.

Reviewer #2: The authors review very important topic for any pediatric diabetologist.

my main concern is in the methodological aspects & results presentation:

1- the process of systematic review entitles essentially duplicate independent review at all stages: title & abstract screening, full text assessment, data extraction. unfortunately the authors only did single review at all stages despite that the total number of articles is considered low which leads to big chance in missing important literature. Please re-do the search according to the proper steps.

2- the authors excluded articles that used parental proxy report. This lead essentially to the exclusion of the young children from the report of the review. I don't see a reason for such an exclusion for an important age group.

3- the authors choose to report on too many outcomes that lead to missing the big picture.

Typically when choosing outcomes the Cochrane recommends not exceeding 7 outcomes. additionally, majority of outcomes are considered a subgroup of interest & I don't see justification to report them separately, and splitting the report based on the type of the scale ( generic vs. DM specific) is not practical.

I would recommend re-doing the analysis and performing meta-analysis based on fewer outcomes pooled via SMD (standardized mean difference) and showing 1 or 2 subgroups:

1- overall quality of life. possible subgroup using type of the scale ( generic vs. DM specific)

2- hypoglycemia fear . possible subgroup using hypoglycmeia type

reporting sub-scales of quality of life is discouraged because those sub-scales in general are used to drive a total score, and to be able to do such analysis the sub-scale will need to be validated for such use.

Heterogeneity in such a review is expected because of the advances in the technology and DM treatment, type of hypoglycemia report, age groups, HbA1c. Therefore, authors are encouraged to explore those factors rather than not doing the meat-analysis.

4- It will be useful to add in appendix type of scales reported in the paper, description of the scale, the cronbach alpha

5- following the adjustments recommended please restructure the results in a comprehensible easy way to read, not all results need to be described. you just need to focus on the main outcomes and leave the other s in the table.

6. PLOS authors have the option to publish the peer review history of their article (what does this mean?). If published, this will include your full peer review and any attached files.

Reviewer #1: No

Reviewer #2: No

---

## [Author Response · Author response to Decision Letter 0]

28 Oct 2021

Responses to reviewer comments:

Comment 1. Date of search was November 2019 (about 2 years ago); authors should update their review to include possible recent publications in 2020 and 2021, such as (Gordon, J., Beresford-Hulme, L., Bennett, H., Tank, A., Edmonds, C., & McEwan, P. (2020). Relationship between hypoglycaemia, body mass index and quality of life among patients with type 1 diabetes: Observations from the DEPICT clinical trial programme. Diabetes, obesity & metabolism, 22(5), 857–865) and (Coolen M, Aalders J, Broadley M, Aanstoot HJ, Hartman E, Hendrieckx C, Nefs G, Pouwer F. Hypoglycaemia and diabetes-specific quality of life in adolescents with type 1 diabetes. Diabet Med. 2021 Aug;38(8):e14565).

 Response 1: We agree that it is important to update our search to include possible recent publications. We have conducted an update of our search, which resulted in 3 extra included articles. We have updated the details in the review accordingly. However, the publication from Gordon et al., 2020 was not included, as this study was conducted in adults with type 1 diabetes aged 18-75 years, and for that reason does not meet the eligibility criteria for inclusion in this review. 

Comment 2: Authors should provide a list of publications assessed for eligibility, with reasons for exclusion of many of them. 

 Response 2: We agree that transparency of the review process is important, therefore we have uploaded the list of papers that were screened during the full text screening stage, with reasons for exclusion. This is added as a supplementary table (Table S1) and referred to in the manuscript on p8, line 160-161. 

Comment 3: Reviewer #1: In data analysis, some terms need to be explained as "diabetes distress"

 Response 3: We agree with the reviewer that it is important to clarify the constructs that are being measured. A recent review defines diabetes distress as ‘the negative emotional or affective experience resulting from the challenge of living with the demands of diabetes’ (1). We have added this definition into the data analysis section, indicated by track changes in the manuscript (p5, line 139-140). 

Comment 4: Regarding the conclusion of the study, why did the authors mentioned that the was insufficient evidence on the relationship between hypoglycemia and QoL despite that we can see from the studies involved in the meta- analysis systematic review that QoL was significantly affected in hypoglycemia.

 Response 4: As we have described in the methods sections, we used the following approach to interpret the evidence and to draw conclusions: “It was determined that there was insufficient evidence to draw a conclusion for an outcome if: a) there were less than three studies examining the association, or b) there was considerable heterogeneity in definitions of hypoglycemia and sample characteristics across studies.” 

 For the relationship between hypoglycemia and generic QoL, there were only three studies with important limitations, such as the use of a non-validated questionnaire to assess QoL, or lack of information on the definition of hypoglycemia. In addition, there was considerable heterogeneity in definitions of hypoglycemia, recall periods and age groups included, among the three studies that examined generic QoL. 

 As we have addressed in the discussion, there was some evidence suggesting that more recent, more frequent, or more severe episodes of hypoglycemia may be associated with lower QoL and related outcomes:

“Although the current evidence suggests no clear association between hypoglycemia and some outcomes, it is important to note that studies were more likely to show statistically significant associations between hypoglycemia and outcomes when SH was experienced recently (in the past 1-3 months), more frequently, or when it involved convulsions, unconsciousness, or coma.”

Comment 5: Reviewer #2: The authors review very important topic for any pediatric diabetologist.

 Response 5: We thank the reviewer for taking the time to read our manuscript and for these kind words.

Comment 6: my main concern is in the methodological aspects & results presentation:

1- the process of systematic review entitles essentially duplicate independent review at all stages: title & abstract screening, full text assessment, data extraction. unfortunately, the authors only did single review at all stages despite that the total number of articles is considered low which leads to big chance in missing important literature. Please re-do the search according to the proper steps.

 Response 6: We agree that it is important to minimize the chances of errors during the review process. We wish to clarify that each step was conducted (in full or in part) by at least 2 people. As also described in the method section of this review, 10% of the abstracts were double screened by a third independent reviewer. Full text screening was conducted by MC (with input from a third reviewer MB), with 10% of the full text records being independently screened by a third reviewer (KM). Data extraction was conducted by MC and KS, and extracted data was checked by two independent reviewers (HC and MB). These steps were taken to minimize errors in the process. In addition, single-screening conducted by experts is methodologically sound (2), so double-screening 10% of the records was deemed to be sufficiently rigorous.

Comment 7: 2- the authors excluded articles that used parental proxy report. This lead essentially to the exclusion of the young children from the report of the review. I don't see a reason for such an exclusion for an important age group.

 Response 7: We partially agree with this reviewer’s comment. As QoL is a highly individual and subjective construct, it is recommended to assess this from the individual’s perspective whenever possible (3, 4). There are inevitable differences between parental proxy report and children’s self-report in terms of QoL, and parents tend to rate their children’s QoL lower than children do, especially in more subjective areas such as emotional and social functioning (3, 5). In addition, parental variables, such as their health, education, socioeconomic status, but also their psychological well-being may influence their proxy report of their child’s quality of life (3). 

 Although parental proxy report can be informative and used to complement self-report, it should only be used as a primary outcome in cases where the child is too young or too ill to self-report (3). With regards to minimum age, it is shown that children from the age of 5 can self-report on their QoL in a valid and reliable way, whenever age-appropriate questionnaires are used (6).

 In the full-text screening of this review, only 2 articles were excluded for the reason of proxy report. In addition, of the 27 studies that are included in this review, five studies also assessed parental proxy report. Of these five studies, three included children under the age of 8 years old. Therefore, we believe that our review still provides an overall picture of children and adolescents with type 1 diabetes across different age groups. 

 The scope for this systematic review was to explore the impact of hypoglycemia on QoL and related outcomes in children and adolescents with type 1 diabetes. While we agree with the reviewer that adding the parental proxy report could complement the child’s self-report, the aim of this review was to explore the concept of QoL as broadly as possible. To avoid including too many outcomes, it was decided to only explore self-report of QoL within the scope of this review. 

Comment 8: 3- the authors choose to report on too many outcomes that lead to missing the big picture.

Typically when choosing outcomes the Cochrane recommends not exceeding 7 outcomes. additionally, majority of outcomes are considered a subgroup of interest & I don't see justification to report them separately, and splitting the report based on the type of the scale (generic vs. DM specific) is not practical.

 Response 8: We thank the reviewer for this comment. While we agree there are several outcomes included in the review, we believe there are clear conceptual and methodological grounds for this. One of the issues in past research is that studies used QoL as a synonym for other constructs, such as diabetes distress or health status. Though these outcomes are related to QoL, these are distinct and unique concepts that cannot be used as a substitute to assess QoL (4). To address this confusion in the literature, and thus to include all possible studies that focus on QoL or related outcomes in this population, the search string was quite broad and related outcomes were not predefined. 

 Generic QoL and diabetes-specific QoL are two different constructs (4). While generic QoL comprises a broad spectrum of different domains of life, diabetes-specific QoL measures refer to the impact of diabetes on QoL and are more sensitive to how specific issues of diabetes treatment and management can impair QoL (7). Results of a systematic review on QoL in children and adolescents with type 1 diabetes indicated that although there was no difference in generic QoL between them and their peers without diabetes, there were specific impacts of diabetes on daily functioning and psychological wellbeing (8). 

 Therefore, there is a need to report generic and diabetes-specific QoL separately, to address these current problems in the literature and to clearly indicate the gaps in the field. The decisions not to include parent proxy-report outcomes and not to conduct meta-analyses, were based on our acknowledgement of the broad scope of outcomes included in the review. 

Comment 9: I would recommend re-doing the analysis and performing meta-analysis based on fewer outcomes pooled via SMD (standardized mean difference) and showing 1 or 2 subgroups:

1- overall quality of life. possible subgroup using type of the scale (generic vs. DM specific)

2- hypoglycemia fear. possible subgroup using hypoglycemia type

 Response 9: We thank the reviewer for this comment. Initially, we opted to do a meta-analysis if the final includes reported sufficient information to do so. However, due to the heterogeneity in outcomes, definitions of hypoglycemia and age groups across studies, a narrative synthesis was the most appropriate approach to synthesis. As suggested in the Cochrane guidelines (9), the research question determines whether a combination of outcomes has a meaningful interpretation. If studies that are too diverse are included in a meta-analysis, differences in effects may be obscured. 

 The purpose of this review was to summarize the evidence for an association between hypoglycemia and a broad range of outcomes (both specifically measuring QoL or its subdomains, and measuring outcomes closely related to, but conceptually distinct from, QoL). The outcomes within this systematic review are too diverse and distinct from one another to warrant a meta-analysis. The “exposure” (hypoglycemia) was also highly heterogeneous in its definition and measurement across studies. If we were to perform a meta-analysis, this could only be done on some outcomes, which would still mean that the majority of the evidence would be summarized in a narrative synthesis. 

 If we followed the suggestion of the reviewer, we would only be comparing three studies, some of which have used different measures. While comparisons are often made between different measures, these are not always valid comparisons. Therefore, we believe that in this systematic review, pooling outcomes together would not result in meaningful findings. 

Comment 10: reporting sub-scales of quality of life is discouraged because those sub-scales in general are used to drive a total score, and to be able to do such analysis the sub-scale will need to be validated for such use.

 Response 10: We hold the opinion that it is of great of interest to report on the subscales separately. The total score is less sensitive to specific impacts, and by only reporting the total scores the impact of QoL on individual domains could be masked. 

 The aim of this review was to explore the impact of hypoglycemia on QoL. Since QoL is comprised of different domains (4), it is relevant in relation to our aim to explore whether some areas might be more (or less) impacted by hypoglycemia. It could be that, for example, there is no impact on global QoL, but there is an impact on some domains of QoL. These subtle differences would not be reflected if we only reported on the total scores. In addition, we believe it is reasonable to use the subscales when there are psychometric properties available for them, which is the case for the scales included in this review (10, 11).

Comment 11: Heterogeneity in such a review is expected because of the advances in the technology and DM treatment, type of hypoglycemia report, age groups, HbA1c. Therefore, authors are encouraged to explore those factors rather than not doing the meat-analysis.

 Response 11: We agree that this is an interesting and relevant question. However, across studies included in this review, 25% did not information on participants treatment regimen, and 85% did not report on glucose monitoring methods. For the studies that did provide this information, analyses between hypoglycemia and QoL or related outcomes were not conducted separately for these groups which precludes the ability to conduct subgroup analyses. While the use of technology has shown promising results for some people with diabetes in relation to minimizing hypoglycemia, many people with diabetes are regularly experiencing hypoglycemia (12). Similarly, studies have shown that HbA1c is an unreliable risk factor for hypoglycemia (13, 14); one can experience hypoglycemia regardless of one’s HbA1c. Since the aim of this review is to explore the association between hypoglycemia and QoL, we believe that, although interesting, these questions are outside of the scope of this review. Additionally, type of hypoglycemia report and age groups are discussed in the discussion section of this paper: 

“Future studies should also use longitudinal /prospective study designs and modern methods, such as continuous glucose monitoring, for a more objective assessment of hypoglycemia that does not rely on recall of episodes, to determine the direct, day-to-day impact of hypoglycemia on various domains of QoL in children and adolescents with type 1 diabetes.”

“The majority of studies included in this review pooled children and adolescents together when examining the link between hypoglycemia and QoL or related outcomes. Although these studies usually included age-appropriate assessments of outcomes, the impact of hypoglycemia on these outcomes might be different for children and adolescents.”

Comment 12: 4- It will be useful to add in appendix type of scales reported in the paper, description of the scale, the Cronbach alpha. 

 Response 12: We thank the reviewer for this suggestion and agree it could be helpful for the reader to have an overview of the scales that are reported in the paper, given the variety of scales used across studies. We have added a supplementary table (Table S2) with an overview of the scales and their subscales and referred to this in the manuscript (p9, line 191-192). As cronbach’s alpha is different for each study in which the scale has been used, it is not possible to give a general alpha for the scale, but we have indicated in the table whether the scales are validated for use in children and adolescents with type 1 diabetes. 

Comment 13: 5- following the adjustments recommended please restructure the results in a comprehensible easy way to read, not all results need to be described. you just need to focus on the main outcomes and leave the others in the table.

 Response 13: We agree that it is important to describe the results in a comprehensible and easy to read way. We also believe it is important to provide the reader with necessary for interpretation of the findings. We have used a consistent approach in the narrative synthesis, to avoid bias in reporting the results. As the aim of this review was to explore the relationship between hypoglycemia and QoL, it is of great importance how these key constructs, (i.e., hypoglycemia and QoL) have been assessed across different studies. This identifies key gaps in the literature that should be addressed in future studies, which are described in the discussion section of this manuscript. In addition, this is in line with items 20c and 20d from the PRISMA Checklist, that suggest all results should be presented. 

 To address the reviewer’s comment, we have taken another look at the results section and have taken out details that were considered less important and were not imperative to interpret findings, indicated by track changes in the manuscript. 

Additional comment from the authors: During the revision process, we found that one of the previously included studies actually met one of our exclusion criteria (Northam et al, 2010, ref #47 in the submitted manuscript) did not meet the inclusion criteria, so this study has been removed from analysis. This results in a total of 27 studies. 

References 

1. Skinner TC, Joensen L, Parkin T. Twenty-five years of diabetes distress research. Diabetic Medicine. 2020;37(3):393-400.

2. Waffenschmidt S, Knelangen M, Sieben W, Bühn S, Pieper D. Single screening versus conventional double screening for study selection in systematic reviews: a methodological systematic review. BMC medical research methodology. 2019;19(1):132.

3. Sherifali D, Pinelli J. Parent as proxy reporting: implications and recommendations for quality of life research. Journal of family nursing. 2007;13(1):83-98.

4. Speight J, Holmes-Truscott E, Hendrieckx C, Skovlund S, Cooke D. Assessing the impact of diabetes on quality of life: what have the past 25 years taught us? Diabetic medicine : a journal of the British Diabetic Association. 2020;37(3):483-92.

5. Eiser C, Morse R. Can parents rate their child's health-related quality of life? Results of a systematic review. Quality of life research : an international journal of quality of life aspects of treatment, care and rehabilitation. 2001;10(4):347-57.

6. Varni JW, Limbers CA, Burwinkle TM. How young can children reliably and validly self-report their health-related quality of life?: an analysis of 8,591 children across age subgroups with the PedsQL 4.0 Generic Core Scales. Health and quality of life outcomes. 2007;5:1.

7. Patrick DL, Deyo RA. Generic and disease-specific measures in assessing health status and quality of life. Medical care. 1989;27(3 Suppl):S217-32.

8. Nieuwesteeg A, Pouwer F, van der Kamp R, van Bakel H, Aanstoot HJ, Hartman E. Quality of life of children with type 1 diabetes: a systematic review. Curr Diabetes Rev. 2012;8(6):434-43.

9. McKenzie J, Brennan S. Chapter 12: Synthesizing and presenting findings using other methods. In: Higgins JPT, Thomas J, Chandler J, Cumpston M, Li T, Page MJ, et al. Cochrane: 2021. Available from: www.training.cochrane.org/handbook. [Accessed 4 January 2021].

10. Varni JW, Burwinkle TM, Jacobs JR, Gottschalk M, Kaufman F, Jones KL. The PedsQL in type 1 and type 2 diabetes: reliability and validity of the Pediatric Quality of Life Inventory Generic Core Scales and type 1 Diabetes Module. Diabetes care. 2003;26(3):631-7.

11. Hullmann SE, Ryan JL, Ramsey RR, Chaney JM, Mullins LL. Measures of general pediatric quality of life: Child Health Questionnaire (CHQ), DISABKIDS Chronic Generic Measure (DCGM), KINDL-R, Pediatric Quality of Life Inventory (PedsQL) 4.0 Generic Core Scales, and Quality of My Life Questionnaire (QoML). Arthritis care & research. 2011;63 Suppl 11:S420-30.

12. Cherubini V, Rabbone I, Lombardo F, Mossetto G, Orsini Federici M, Nicolucci A. Incidence of severe hypoglycemia and possible associated factors in pediatric patients with type 1 diabetes in the real-life, post-Diabetes Control and Complications Trial setting: A systematic review. Pediatr Diabetes. 2019;20(6):678-92.

13. Karges B, Kapellen T, Wagner VM, Steigleder-Schweiger C, Karges W, Holl RW, et al. Glycated hemoglobin A1c as a risk factor for severe hypoglycemia in pediatric type 1 diabetes. Pediatric diabetes. 2017;18(1):51-8.

14. Cooper MN, O'Connell SM, Davis EA, Jones TW. A population-based study of risk factors for severe hypoglycaemia in a contemporary cohort of childhood-onset type 1 diabetes. Diabetologia. 2013;56(10):2164-70.

---

## [Decision Letter · Decision Letter 1]

19 Nov 2021

The impact of hypoglycemia on quality of life and related outcomes in children and adolescents with type 1 diabetes: a systematic review

PONE-D-21-24300R1

Dear Dr. Coolen,

We’re pleased to inform you that your manuscript has been judged scientifically suitable for publication and will be formally accepted for publication once it meets all outstanding technical requirements.

Kind regards,

Elsayed Abdelkreem, MD, PhD

Academic Editor

PLOS ONE

Additional Editor Comments (optional):

Reviewers' comments:

Reviewer's Responses to Questions

**Comments to the Author**

1. If the authors have adequately addressed your comments raised in a previous round of review and you feel that this manuscript is now acceptable for publication, you may indicate that here to bypass the “Comments to the Author” section, enter your conflict of interest statement in the “Confidential to Editor” section, and submit your "Accept" recommendation.

Reviewer #1: All comments have been addressed

2. Is the manuscript technically sound, and do the data support the conclusions?

Reviewer #1: Partly

3. Has the statistical analysis been performed appropriately and rigorously? 

Reviewer #1: N/A

4. Have the authors made all data underlying the findings in their manuscript fully available?

Reviewer #1: Yes

5. Is the manuscript presented in an intelligible fashion and written in standard English?

Reviewer #1: Yes

6. Review Comments to the Author

Reviewer #1: Authors have addressed most of the reviewers' comments in their response.

English editing is still required

7. PLOS authors have the option to publish the peer review history of their article (what does this mean?). If published, this will include your full peer review and any attached files.

Reviewer #1: No

---

## [Editor Report · Acceptance letter]

23 Nov 2021

PONE-D-21-24300R1 

The impact of hypoglycemia on quality of life and related outcomes in children and adolescents with type 1 diabetes: a systematic review 

Dear Dr. Coolen:

I'm pleased to inform you that your manuscript has been deemed suitable for publication in PLOS ONE. Congratulations! Your manuscript is now with our production department. 

Kind regards, 

on behalf of

Dr. Elsayed Abdelkreem 

Academic Editor

PLOS ONE